# *Arabidopsis* PII Proteins Form Characteristic Foci in Chloroplasts Indicating Novel Properties in Protein Interaction and Degradation

**DOI:** 10.3390/ijms222312666

**Published:** 2021-11-23

**Authors:** Natalie Krieger, Kai-Florian Pastryk, Karl Forchhammer, Üner Kolukisaoglu

**Affiliations:** 1Center for Molecular Biology of Plants (ZMBP), University of Tübingen, Auf der Morgenstelle 32, 72076 Tübingen, Germany; natalie.krieger@zmbp.uni-tuebingen.de (N.K.); kai-florian.pastryk@student.uni-tuebingen.de (K.-F.P.); 2Interfaculty Institute of Microbiology and Infection Medicine, University Tübingen (IMIT), Auf der Morgenstelle 28, 72076 Tübingen, Germany

**Keywords:** plant PII protein, protein–protein interaction, PII foci, BiFC, FRET/FLIM, plastidic protein degradation, cpUPR

## Abstract

The PII protein is an evolutionary, highly conserved regulatory protein found in both bacteria and higher plants. In bacteria, it modulates the activity of several enzymes, transporters, and regulatory factors by interacting with them and thereby regulating important metabolic hubs, such as carbon/nitrogen homeostasis. More than two decades ago, the PII protein was characterized for the first time in plants, but its physiological role is still not sufficiently resolved. To gain more insights into the function of this protein, we investigated the interaction behavior of *At*PII with candidate proteins by BiFC and FRET/FLIM *in planta* and with GFP/RFP traps in vitro. In the course of these studies, we found that *At*PII interacts in chloroplasts with itself as well as with known interactors such as N-acetyl-L-glutamate kinase (NAGK) in dot-like aggregates, which we named PII foci. In these novel protein aggregates, *At*PII also interacts with yet unknown partners, which are known to be involved in plastidic protein degradation. Further studies revealed that the C-terminal component of *At*PII is crucial for the formation of PII foci. Altogether, the discovery and description of PII foci indicate a novel mode of interaction between PII proteins and other proteins in plants. These findings may represent a new starting point for the elucidation of physiological functions of PII proteins in plants.

## 1. Introduction

PII signaling proteins are ubiquitously distributed in all prokaryotes and have been conserved in the evolution of the “green lineage”. Therefore, PII proteins are found in all plants, from cryptogams to angiosperms, where they are almost exclusively localized in the plastids. In prokaryotes, PII proteins are widely distributed in bacteria and many archaea [1,2]. These trimeric proteins have a highly conserved structure and regulate their targets through tight protein–protein interactions, mainly mediated through a flexible, solvent loop structure of about 18 amino acids, known as the so-called T-loop. The T-loop can adopt various conformations depending on the effector molecules *At*P, ADP, or Mg-ATP plus 2-oxoglutarate bound in the three effector binding sites, which are located in the clefts between the subunits. Therefore, the levels of the carbon/nitrogen status-reporter 2-oxoglutarate as well as the energy state, sensed by the *At*P to ADP ratio, are integrated by PII. This allows PII to control a multitude of cellular functions, mainly related to nitrogen assimilation, but also to central carbon flux and other core features of metabolism such as NAD-synthesis [3,4].

In cyanobacteria, which are the phylogenetic ancestors of chloroplasts of the plant kingdom through endosymbiosis, PII signaling has been investigated in detail [5]. The signaling principles are highly conserved compared to heterotrophic bacteria; however, cyanobacteria appear to have evolved specific PII regulatory targets. The controlling enzyme of the arginine pathway, N-acetyl-L-glutamate kinase (NAGK), appears to be a dominant target in these organisms, although recent studies also revealed PII-NAGK interaction in non-photosynthetic bacteria [4]. Moreover, in cyanobacteria, PII controls the flux of newly fixed carbon by controlling a protein that acts as the inhibitor of phosphoglycerate mutase (PGAM) [6].

In 1998, the first plant PII protein was identified and characterized from *Arabidopsis*. Surprisingly, the amino acid sequence revealed an identity of 50% or more to homologous protein sequences from *E. coli* or cyanobacteria [7]. Due to the degree of conservation of PII proteins from bacteria to plants, similar regulatory mechanisms as described above were expected to be mediated by these proteins in plants. Transcription of the corresponding *At*PII gene revealed it to be under the control of carbohydrates and nitrogen, indicating a regulatory role of *At*PII in C/N homeostasis. In further studies, it was reported that plant PII proteins also bind to the NAGK enzyme as in bacteria [8,9]. This interaction has been shown to activate NAGK enzyme activity and to be strictly regulated by different metabolites, such as *At*P, 2-oxoglutarate, or glutamine [10,11,12,13]. The glutamine dependence was identified in the green alga *Chlamydomonas reinhardtii* [13] and was resolved to be due to a glutamine-binding C-terminal extension of plant PII proteins. Strikingly, this extension is modified in *Arabidopsis thaliana* in such a way that *At*PII binds NAGK in a glutamine-independent manner. The evolution of glutamine sensing by plant PII proteins was recently reviewed in [14].

Additionally, the interaction of Biotin Carboxyl Carrier Proteins (BCCPs) has been shown for plant PII proteins [12], indicating their regulatory role in fatty acid biosynthesis, which was later found to also occur in bacteria [15,16].

Nevertheless, major physiological phenomena in plants under the control of PII proteins are still waiting to be unraveled. Arabidopsis plants overexpressing *At*PII showed enhanced anthocyanin accumulation by glutamine application [7]. This led to the hypothesis of PII in plants as a hub for C/N balance, which was supported by the finding of direct glutamine binding of these proteins [13]. In contrast, the phenotype of PII overexpressing plants was unexpectedly moderate. Homologous overexpression of PII in *Lotus japonicus* led to the deregulation of polyamine content and nodule number under high nitrogen supply [17] as well as the reduction in water loss due to altered stomatal opening [18], but no drastic alteration of plant phenotype has been reported. Furthermore, knock-out and knock-down *At*PII plants were phenotypically slightly affected [19]. The utilization of *At*PII mutants revealed reduced contents of arginine biosynthesis metabolites under certain nitrogen supply conditions [20] and an increased uptake of nitrite in chloroplasts [21].

Altogether, these findings led to the conclusion that PII is not crucial for plant nitrogen sensing [22]. Instead, it turned out that several proteins, such as the TOR (Target of Rapamycin) kinase, the GCN2 protein kinase, GLRs (glutamate receptor-like proteins), or several other candidates, may have taken over the task of nitrogen sensing in plants (for summaries see [10,22]). Among them, the TOR signaling pathway appears to play a central integrative function in this respect [23]. It has been speculated that PII is just responsible for the upregulation of arginine biosynthesis under high glutamine supply [22]. In this case, the question about the function of PII proteins in *Arabidopsis* and other Brassicaceae would remain, as these proteins had lost their glutamine binding moiety [13].

Moreover, the variety of PII interaction partners in bacteria and plants and the different metabolic pathways regulated by PII in bacteria, together with the degree of PII conservation during evolution, led to the question of whether this protein also plays a more pronounced role in plants. In the present study, we addressed this question by studying the interaction partners and functions of the PII protein in *Arabidopsis*. During these studies, we observed a specific aggregation behavior of this protein in plastids, leading to the discovery of novel interaction partners of *At*PII within this suborganellar structure. This specific aggregation behavior pointed to a central role for plant PII proteins in a novel context. Furthermore, we were interested in the mechanism leading to this aggregation, which seems to be mediated by the C-terminal component of the protein.

## 2. Results

### 2.1. At*PII* Tagged with Fluorescent Proteins Forms Dot-Like Aggregates in Chloroplasts

The first attempts to analyze the subcellular localization of *At*PII revealed that GFP-tagged versions of this protein under the control of the 35S CaMV-promotor (*p35S CaMV::At*PII*_cDNA_-GFP*) were found in chloroplasts of transiently transfected tobacco cells. Most interestingly, they appeared as roundly shaped, dot-like or focal aggregates of varying sizes (Figure 1A). To exclude the possibility that these aggregates evolved due to overexpression phenomena, a GFP fusion of the genomic PII sequence driven by its endogenous promotor (*pAt*PII*::At*PII*_genomic_-GFP*) was introduced into tobacco cells. The expression of this construct also showed plastidial aggregation (Figure 1B). Furthermore, the same construct was used for the stable transformation of *Arabidopsis thaliana* Col-0 plants. Additionally, in the cells of these transformed plants, *At*PII-GFP aggregated in foci in chloroplasts, as observed previously in tobacco cells (Figure 1C), which shows that this phenomenon was not caused by heterologous overexpression. As we observed this focal aggregation of PII proteins regularly, which can be seen in the course of this report, we coined this phenomenon as PII foci.

The specific subcellular localization pattern of *At*PII raised the question of whether the formation of PII foci is a general phenomenon or if it is restricted to specific conditions. Therefore, we transformed an *At*PII genomic sequence that encoded a C-terminal GFP fusion and was under the control of an ubiquitin promotor (*pUBQ10::At*PII**_genomic_*-GFP*) into *Arabidopsis thaliana* Col-0 to observe *At*PII localization under different conditions. We then tested the impact of different temperature and light regimes on *At*PII localization in this line. As can be seen in Figure 2, differences in the formation of PII foci can be observed depending on temperature or changing light quality. In this experiment, most chloroplasts displayed GFP fluorescence throughout the organelle. Nevertheless, PII foci of different sizes formed under all tested conditions to different extents without a clear recognizable trend (Figure 2).

### 2.2. At*PII* Interaction with Itself and Other Proteins Takes Place in PII Foci

Next, we were interested in whether PII foci would also appear when *At*PII interacts with itself or with other proteins. Therefore, we applied bimolecular fluorescence complementation (BiFC) to analyze the physical interaction of the candidate proteins. As shown in Figure 3A, co-expression of *At*PII fused with nYFP and cYFP in tobacco led to dot-like fluorescence, as observed beforehand. As *At*NAGK and *At*BCCP1 were previously characterized as PII interactors, we also cloned both coding sequences into the same 2in1 BiFC vector together with *At*PII. Moreover, in these cases, focal aggregation of fluorescence appeared after infiltration (Figure 3B,C).

These observations implied that the interaction of *At*PII with itself and other proteins occurs in PII foci. To confirm this finding, we co-expressed *At*PII and its putative interacting proteins as C-terminal fusions to GFP and mCherry, respectively, in suitable 2in1 vectors. Then, these vectors were infiltrated into tobacco and FRET/FLIM analyses were applied to co-localizing fluorescent signals. In Figure 4A–C, the co-localization of GFP and mCherry signals was observed in all experiments in dot-like structures. Fluorescence lifetime measurements (FLIM) revealed significantly reduced GFP fluorescence lifetime values of the co-localizing signals in comparison to *At*PII-GFP alone (Figure 4D). This observation confirms the assumption of physical interaction of *At*PII with itself and other proteins in PII foci as such a reduction in fluorescence lifetime only occurs when two fluorescent proteins are at a critical distance of <10 nm [25].

Further analysis of *At*PII-GFP together with *At*NAGK-mCherry additionally revealed that the formation of PII foci evolves within seconds and that they only persist temporarily. In Figure 5, a series of images of such an interaction over a time range of more than 2 min is shown. As can be seen in this figure, some of the PII foci evolved and vanished within a minute whereas others were visible over the whole range of time. Further analyses of PII foci showed that some of them were even stable for more than half an hour (Appendix A).

### 2.3. Characterization of PII Foci Points to Plastidic Protein Degradation

Detecting *At*PII in focal aggregates in plastids opened the question of whether these suborganellar structures have been characterized before. As one candidate structure in plastids, we tested whether PII foci are part of nucleoids. Therefore, DNA in the chloroplasts was visualized with YO-PRO1™ iodide, as previously described [26]. In all the tobacco cells expressing *At*PII-RFP, this fluorescent signal could be clearly distinguished from that of YO-PRO1™ iodide (Appendix A), which indicates that PII foci are not part of the nucleoids in plastids.

Subplastidial compartments, similar to PII foci, have also been reported to represent vesicle-like structures for protein degradation from chloroplasts [27,28]. The small subunit of Rubisco (RBCS) is known to be part of such plastidial protein degradation vesicles, known as the Rubisco containing bodies (RCBs) [27,29,30].

To test whether PII foci are part of the plastidic protein degradation apparatus, we co-expressed *At*PII-GFP with *At*RBCS3B-RFP in tobacco cells. Co-localization of the fluorescent signals in chloroplasts (Figure 6A), as well as the decreased GFP lifetime in FLIM analyses (Figure 6D), confirmed our assumption. Furthermore, some of the co-localizing fluorescent foci could be found outside the chloroplasts (Figure 6A), supporting the assumption that *At*PII is part of plastidial protein degradation vesicles. To support this hypothesis, we also tested other proteins involved in autophagy-dependent protein degradation of RCBs [27,31]. For this purpose, TagRFP or RFP-tagged *At*NBR1, *At*g8e, and *At*g8g from *A. thaliana* were co-expressed with *At*PII-GFP [32]. In particular cases, the co-localization of GFP and RFP signals was also observed (Appendix A).

The co-localization of *At*PII with proteins mediating different protein degradation pathways led to the assumption that *At*PII may even be involved in the earlier steps of this process, such as protein quality control (PQC). Previously, it was shown that two enzymes of the methylerythritol-4-phosphate (MEP) pathway, deoxyxylulose-5-phosphate synthase (DXS) and reductoisomerase (DXR), undergo such PQC in *Arabidopsis* chloroplasts by aggregation, similar to PII foci [33,34]. To test whether *At*PII is also involved in PQC, such as *At*DXR and *At*DXS, they were also cloned into suitable 2in1 vectors together with *At*PII for fluorescence microscopy and subsequent FRET/FLIM analyses in tobacco after infiltration. The microscopic images revealed partial co-localization of *At*DXR and *At*DXS with PII foci (Figure 6B,C). Additionally, in FRET/FLIM analyses of co-localizing fluorescent signals a significant decrease in fluorescent lifetime was detected, indicating in vivo physical interaction of *At*PII with both *At*DXR and *At*DXS (Figure 6D).

### 2.4. The C-Terminus of At*PII* Is Responsible for the Formation of PII Foci

The observation that PII is found in plastidic protein degradation aggregates opened the question of whether this protein is only found there due to co-degradation in PQC, or whether PII is indeed mediating the aggregation with other proteins. Furthermore, we wondered about the differing sizes of the PII foci that we observed in our microscopic studies. We assumed that a targeted change in PII protein structure could lead to differences in PII foci formation. To test this hypothesis, we created a C-terminally truncated version of *At*PII without the so-called Q-loop. Almost all plant PII proteins possess a Q-loop motif at their C-terminus, which is responsible for glutamine binding. The Q-loop affects the conformation of the T-loop in plant PII proteins, which is responsible for interaction with target proteins [13].

Therefore, an additional version of *At*PII*-GFP* without the Q-loop was cloned, in which the last 15 amino acids (*At*PII*_ΔCT15_-GFP*) were deleted. For further analyses we co-expressed *At*PII-GFP or *At*PII_ΔCT15_-GFP with *At*PII-mCherry, *At*NAGK-mCherry, and *At*RBCS3B-mCherry into tobacco leaves. To test the physical interaction of *At*PII-GFP and *At*PII_ΔCT15_-GFP with the mCherry-tagged interactors, the GFP-tagged proteins were first isolated from protein extracts using GFP traps. Afterwards, the co-eluted proteins were detected by Western blot analysis with an αRFP antibody. As shown in Appendix A, infiltration of either *At*PII-GFP or *At*PII_ΔCT15_-GFP led to the co-purification of all co-infiltrated interactors. These results indicate that *At*PII_ΔCT15_-GFP is still able to interact with itself and its binding partners.

In contrast, fluorescent microscopic analyses of these co-infiltrations revealed differences between the *At*PII variants in terms of PII foci formation. Co-infiltrated with *At*PII-GFP, the number of PII foci was higher and their size smaller in comparison to *At*PII_ΔCT15_-GFP (Figure 7A–F). Additionally, with *At*PII_ΔCT15_-GFP, the GFP signals were not confined to PII foci, but tended to spread within the whole chloroplast (Figure 7D–F).

This observation was supported by quantitative analysis of these images. Therefore, we differentiated and counted the co-localizing fluorescence signals according to their size in signals smaller than 1µm and those larger. In Figure 7G–I, the number of co-localizing fluorescence signals per plastid is provided. It is obvious that the truncation of the C-terminus of *At*PII leads to a significant reduction in PII foci number in all tested cases. Specifically, the number of small signals decreased significantly in chloroplasts expressing *At*PII_ΔCT15_-GFP, whereas the number of large signals did not differ significantly (Figure 7G–I).

## 3. Discussion

In the present study it was shown that *At*PII, either alone or co-expressed with its interaction partners, was almost exclusively found in aggregates in the chloroplasts. This aggregation, coined as PII foci in this study, took place irrespective of the promotor driving *At*PII expression or the plant species (*Arabidopsis* or *Nicotiana*) it was expressed in (Figure 1 and Figure 2). This impression was supported by the microscopic images of BiFC experiments of *At*PII interaction with itself, where the fluorescent signals were almost strictly confined to foci (Figure 3A).

As the physiological function of the PII protein in plants is still not clearly resolved, the appearance of such a characteristic structure as the PII foci raised several questions. PII proteins were characterized in plants for the first time more than 20 years ago [7]. An obvious question regarding PII foci is whether this phenomenon was observed before that time. Revisiting former localization studies of plant PII proteins revealed that the aggregation of this protein in plastids had indeed been observed before but did not attract further interest. In *Arabidopsis* cells, PII was detected for the first time in chloroplasts by immunolocalization in a dotted pattern [35]. GFP fusions of two PII isoforms from maritime pine (*Pinus pinaster*) expressed in tobacco cells also revealed a similar localization pattern [36]. Focal aggregation of PII has even been observed in evolutionary distant organisms such as the cyanobacterium *Synechococcus elongatus* [37]. These reports indicate that the formation of PII foci is an evolutionary conserved phenomenon and is not limited to *At*PII due to the slightly aberrant structure of the C-terminus without the glutamine binding site [13].

The presented results from our FRET/FLIM analyses, BiFC, and co-immunoprecipitation experiments strongly support the direct interaction between *At*PII and the tested proteins within PII foci. However, final evidence for the physical interaction of *At*PII has only been provided for *At*NAGK [9,38], whereas physical binding analyses with methods such as surface plasmon resonance (SPR), isothermal calorimetry (ITC), or microscale thermophoresis (MST) for all other binding partners are still lacking. However, the type of interaction may still be indirect, as it has been shown for bacterial PII–PipX complexes to interact with the transcription factor PlmA [39], although a PipX-like protein has not been shown in plants to date. Nevertheless, the results of our studies revealed the close proximity (approximately <10 nm) between *At*PII and the tested proteins. These findings make a regulatory influence of *At*PII on neighboring proteins very likely. These proteins in close proximity of *At*PII and their functions will be the next target of future studies.

Our experiments with *At*PII_ΔCT15_-GFP trap analysis indicated that the C-terminal Q-loop does not seem to be essential for the binding of PII to its interactors, although it has been suggested that this region stabilizes the T-loop of *At*PII and thereby the binding to *At*NAGK [13]. This contradiction could be explained by the different sensitivity range of the analytics, where biophysical methods such as SPR fail to detect weaker interactions [40]. The fact that the formation of PII foci requires the Q-loop indicates that PII-target complex formation may not be the driving force in foci formation, but rather indicates a regulatory role of the C-terminus of *At*PII in the formation of PII foci. This region was shown to be responsible for glutamine binding of plant PII protein with an evolutionary peculiar exception of its homologs in Brassicaceae [13]. In this regard, it would be highly interesting to repeat these experiments with PII homologs from other plants, especially with C-terminally truncated versions, to investigate whether there is a general correlation between glutamine, nitrogen supply, and the formation of PII foci. In addition, the aggregation and disintegration dynamics of PII foci (Figure 5 and Appendix A) in response to light, temperature, and nutrient supply should be analyzed in more detail; this may provide further valuable information about the role of PII proteins in these processes.

The observation of PII foci leads to the greater question of whether this structural feature can be related to a physiological function. As described in Section 2.3, one putative function of PII in plants may be its contribution to plastidic protein degradation. This was deduced due to its subcellular co-localization and interaction with different components of the plastidic protein degradation pathways (Appendix A) and the observation of PII foci outside of chloroplasts (Figure 6A,B). Further analyses indicated the physical interaction of *At*PII in PII foci with *At*DXR and *At*DXS, which are known to aggregate during inactivation and degradation [33,34]. The accumulation and aggregation of both proteins is initiated and regulated by the so-called chloroplast unfolded protein response (cpUPR). In this process, different Clp proteases and heat shock proteins (HSPs) are employed to regulate the levels of proteins, such as *At*DXR and *At*DXS [34]. Furthermore, pharmacological and genetic approaches have revealed a crucial role of plastome gene expression (PGE) in cpUPR [34]. Our results of *At*PII interacting with *At*DXR and *At*DXS in PII foci imply the involvement of PII in cpUPR. This assumption is supported by the experiments with the C-terminally truncated *At*PII variant *At*PII_ΔCT15_-GFP. Infiltration of this construct together with *At*PII-, *At*NAGK-, or *At*RBCS3B-mCherry, led to the diffusion of PII foci (Figure 7. This is reminiscent of images of *At*DXS-GFP expressing *Arabidopsis* plants either treated with an inhibitor of PGE or mutated in plastidic ribosome formation. In both cases, the dotted aggregation pattern of *At*DXS-GFP was dispersed [34]. A putative function of PII in cpUPR would explain why, so far, the role of this protein in plants remains largely cryptic. Several different anabolic pathways such as the synthesis of tetrapyrrole, chlorophyll, carotenoids, and isoprenoids, have been identified as prone to regulation of protein aggregation and degradation by cpUPR (for a summary, see [41]). These pathways were not the focus of the investigation of PII-regulated processes, yet they should be investigated in the future.

It is noteworthy that in this study several novel interactors of a PII protein from plants were reported. In addition to well characterized interactors such as *At*NAGK, *At*BCCP1 and -2, and *At*BADC1-3 [12], the number of proteins and their functional range seem to be growing. This resembles the situation found for bacterial PII homologs and their interaction partners: for the PII homolog GlnZ from *Azospirillum brasilense,* 37 interaction partners could be identified in ligand fishing assays [4]. In the cyanobacterium *Synechocystis* sp., all major transporter proteins involved in ammonium, nitrate, and urea transport interact with the bacterial PII homolog GlnB [42]. Most interestingly, only NAGK could be identified as a common interactor in both studies, whereas all other proteins were functionally different [4]. Therefore, it seems reasonable that the list of this network in bacteria cannot be closed soon. With the novel *Arabidopsis* PII interactors reported here, it can be concluded that PII interaction networks with many diverse protein partners are also an evolutionary conserved property in higher plants. However, these novel interactors escaped identification by co-purification experiments due to moderate affinities [12], and therefore should be analyzed in more detail.

Nevertheless, even the characterized PII network in plants implies multiple regulatory functions of plant PII proteins. Interestingly, localization studies of *At*NAGK alone also revealed a dotted distribution in *Arabidopsis* chloroplasts such as PII foci [43]. The same was observed for *At*BADC1, a protein that facilitates the physical interaction of the PII interactor *At*BCCP1 within htACCase [44]. It was further mentioned in this report that several other proteins from *Physcomitrella* and potato involved in lipid metabolism were also observed in foci in chloroplasts [45,46]. Furthermore, colleagues noticed a similar subcellular distribution of PII proteins [44]. In this regard, plastidic enzymes of lipid metabolism may be additional candidates of the PII foci network, which needs to be confirmed in future.

Regardless of the physiological functions of PII foci, the question remains as to whether PII is central to these suborganellar structures. Therefore, many of the presented subcellular localization studies have to be repeated in *Arabidopsis* in an *At*PII null background such as PIIS2, an *At*PII knock-out line that had been isolated and previously described [19]. In the present study, most of the results with *At*PII were achieved in a heterologous system (*N. benthamiana*) with an endogenous PII encoding gene, whose impact on different regulatory phenomena must not be neglected as it can be performed in a null background system. It would be highly interesting to express proteins such as NAGK or BCCP from *Arabidopsis* alone and in conjunction with *At*PII in an *At*PII knock-out line to determine whether the focal aggregation of PII interactors is directly dependent on PII in plants.

Altogether, it can be stated that the presented results unraveled novel interactors of *At*PII proteins, which led to novel physiological processes regulated by PII proteins in plants. Although the regulatory mechanism is still unclear, the aggregation of *At*PII together with its interactors point to different possibilities of regulatory functions; the regulatory role of PII in plastidic protein degradation has to be considered as one possibility. However, the presented data regarding aggregation dynamics and the proportion of extraplastidic PII foci make an exclusive role in this respect less probable. Another possibility to be tested in this regard should be the role of plant PII proteins in the formation of multi-enzyme assemblies or metabolons, which are known to be central to substrate channeling and metabolic regulation in plants too (for a summary, see [47]). As PII proteins bind to a variety of metabolic enzymes, as mentioned above, the involvement of PII as a scaffolding protein is possible. Nevertheless, finding answers to these questions will be one of the central tasks for the elucidation of the physiological functions of PII proteins in plants.

## 4. Materials and Methods

### 4.1. Plant Material and Growth Conditions

For transformation of *Arabidopsis thaliana* (Col-0), plants were grown on T- and R-soil mixed with sand (10:10:1) under long day conditions (16 h light at 18 °C, 8 h dark at 15 °C) at a humidity of 55–60% in a greenhouse. Stable transformed lines were generated using the floral dipping method according to [48].

Syringe-mediated infiltration [49,50] was used for transient transformation of leaves of three- to four-week-old *N. benthamiana* with *A. tumefaciens* carrying plasmids of interest. Growth of *A. tumefaciens* and infiltration was performed according to protocol described in [51] derived from protocols of [49,50,52,53] with the modification that cells were not washed with sterile H_2_O before resuspension in AS medium.

For specific light and temperature treatment of *Arabidopsis*, plants seeds were sown on ½ MS media (Murashige and Skoog basal salt, DUCHEFA Biochemie B. V. (Haarlem, Netherlands)). After stratification for one night at 4 °C, plates were transferred for one day to constant light at 23 °C and placed in black boxes for additional three days under constant light at 22 °C. One plate per condition was placed for 24 h for temperature treatments in dark at 8 °C, 23 °C, and 37 °C, and for light treatments in blue light (BL), green light (GL), red light (RL), and far-red light (FRL). For further microscopic analysis, harvested seedlings were pre-fixed in 2 × SSC + 4% formaldehyde for 4 h, followed by vacuum infiltration three times for 15 sec, and an additional incubation step for 30 min. Seedlings were transferred to 6-well plates and washed once in 2 × SSC overnight and twice for 1 h. Seedlings were mounted on dH_2_O on microscope slides and covered with cover glasses. Microscopic analysis was performed with Zeiss LSM880. Growth conditions and light treatment were modified according to protocol by [54].

### 4.2. Generation of Plant Expression Vectors

Coding DNA sequences (CDS) of *At*PII (*At*GLB1-Start; *At*GLB1-End), *At*PII_ΔCT15_ (AtGLB1-Start; *At*PII-C2A), *At*NAGK (NK_AtNAGKstart; NK_AtNAGKend), *At*BCCP1 (NK_AtBCCP1start-2; NK-BCCP1end), and *At*RBCS3B (NK_RGCS1A-FP; NK_RGCS1A-RP) were amplified from cDNA of *Arabidopsis thaliana* Col-0 seedlings (for primer sequences, see Appendix A) for cloning into pENTR™/D-TOPO^®®^ (pENTR™ Directional TOPO^®®^ Cloning Kits from Invitrogen (Carlsbad, CA, USA)) followed by LR (LR clonase, Invitrogen (Carlsbad, CA, USA)) into either pUBQ10-Dest [55], pH7FWG2,0-Dest, or pB7RWG2,0-Dest ([56]; for specifications of vectors see Appendix A).

Genomic constructs of the endogenous *At*PII promoter (−269 bp), together with the genomic coding sequence of *At*PII or the genomic coding *At*PII sequence only, were amplified using NK_pro*At*PIIstart/AtGLB1-End or *At*GLB1-Start/*At*GLB1-End, respectively, on genomic DNA extracted from *Arabidopsis thaliana* Col-0, followed by cloning into pENTR™/D-TOPO^®®^. These entry constructs were cloned into pMDC107 [57] or pUBQ10-Dest [55], respectively, by LR clonase reaction.

For the generation of 2in1 constructs for BiFC and FLIM analysis, CDS of genes harbouring either P3P2 or P1P4 attachment sites were amplified, followed by BP reaction (BP clonase, Invitrogen (Carlsbad, CA, USA)) into pDONR221-P3P2 and pDONR221-P1P4 (Invitrogen (Carlsbad, CA, USA)), respectively, and LR reaction into pBiFCt-2in1-CC and pFRETgc-2in1-CC. The following primer combinations were used (for primer sequences, see Appendix A): *At*PII P2P3 (NK_attP2P3-PIIstart; NK_attP2P3-PIIend), *At*PII P1P4 (NK_attP1P4-PIIstart; NK_attP1P4-PIIend), *At*NAGK P1P4 (NK_attP1P4-NAGKstart; NK_attP1P4-NAGKend), *At*BCCP1 P1P4 (NK_attP1P4-BCCP1start; NK_attP1P4-BCCP1end), *At*RBCS3B (NK_RGCS1A-P1P4-FP; NK_RGCS1A-P1P4-RP), *At*DXS (NK_attP1-FP-DXS; NK_attP4-RP-DXS), and *At*DXR (NK_attP1-FP-DXR; NK_attP4-RP-DXR). pENTR-L1-GentR-L4 was used for the generation of the donor-only controls for BiFC and FLIM analysis by multisite LR in pBiFCt-2in1-CC and pFRETgc-2in1-CC together with pENTR-L3L2-PIIend.

### 4.3. Microscopic Analyses

Fluorophores were imaged using a Leica TCS SP8 AOBS FLIM and Zeiss LSM880 Airyscan with a 63X/NA1.2 water objective. GFP and YO-PRO™-1 iodide were excited at 488 nm and YFP at 514 nm using an Argon laser. RFP, mCherry, and chlorophyll were excited at 561 nm using a DPSS 561 nm laser.

YO-PRO™-1 iodide (Thermo Fisher Scientific (Waltham, MA, USA)) staining was modified from [26]. *N. benthamiana* leaves were transiently transformed with *A. tumefaciens* harboring *At*PII-RFP under the control of *p35S*, together with *A. tumefaciens* harboring P19 or *A. tumefaciens* harboring P19 alone. Leaf disks were cut out three days after infiltration, incubated in 2 × SSC (0.3 M NaCl, 30 mM sodium citrate, pH 7.0) +/− RNase A (10 μg/mL), or 2 × SSC + 1 × DNase I buffer + DNase I (100 U/mL) for 4 h at 37 °C, and transferred to 2 × SSC + 4% formaldehyde. Leaf disks were vacuum infiltrated for 15 sec at 300 mbar three times, followed by incubation for 15 min. Washing was performed three times in 2 × SSC. Leaf disks were stained with DAPI (1 µg/mL) + YO-PRO™-1 iodide (5 μg/mL) in 2 × SSC overnight and washed 1 × with 1 × SSC. Leaf disks were mounted on microscopic slides on SSC/glycerol (50% 2 × SSC + 50% glycerol) and covered with cover slips. Confocal imaging was performed using a Zeiss LSM880 Airyscan.

BiFC analyses were performed according to a modified protocol of [58] three days after transient transformation of *Nicotiana benthamiana* leaves with *Agrobacterium tumefaciens* harboring 2in1 *pBiFC* vectors of interest. Internal RFP fluorescence was used as the transformation control.

FRET-FLIM measurements were performed with Leica TCS SP8 AOBS FLIM equipped with SymPhoTime software (PicoQuant GmbH (Berlin, Germany)), with PicoHarp 300 (PicoQuant GmbH (Berlin, Germany)) for FLIM measurements and TimeHarp 260 Nano (PicoQuant GmbH (Berlin, Germany)) for rapidFLIM measurements. Measurements were taken of transiently transformed leaf disks of *N. benthamiana* leaves with *A. tumefaciens* GV3101 carrying 2in1 *pFRET* vectors of interest. FLIM measurements were performed according to [59] and [51]. Two biological replicates (FLIM and rapidFLIM) were measured in five to six regions (FLIM) and four to five regions (rapidFLIM) containing plastids in the epidermis. For FLIM measurements, acquisition was performed until 700 photons in the brightest point were counted. For rapidFLIM measurements, acquisition was performed until 1500 photons at the brightest point were counted.

All images acquired were processed using Fiji (Fiji is the same as ImageJ; [60] was based on ImageJ) and Microsoft Office 2019 PowerPoint (Microsoft Corporation (Redmond, Washington, DC, USA).

### 4.4. Statistical Analyses

Statistical analyses of FLIM measurements using two-sided Student’s *t*-test as well as a generation of box plots were carried out in Matlab (The MathWorks Inc. (Natick, MA, USA)). Statistical analyses of PII foci numbers were performed using two-sided Student’s *t*-test in Microsoft Excel.

## Figures and Tables

**Figure 1 ijms-22-12666-f001:**
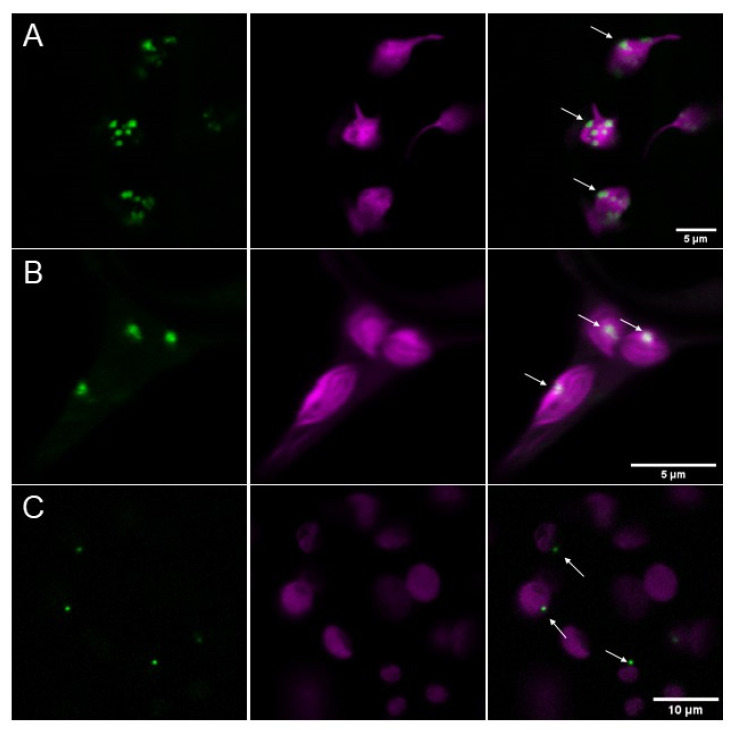
*At*PII aggregates in focal structures in chloroplasts. (**A**) *At*PII-GFP (green) under the control of *p35S* (*p35S CaMV*::*At*PII_cDNA_-*GFP*) and co-expressed with mCherry-tagged transit peptide of tobacco Rubisco (CD3-999 pt-rk [24]; magenta) localizes to plastids in transiently transformed *N. benthamiana* 2 days after infiltration. (**B**) Genomic *At*PII-GFP (green) expressed under the control of the endogenous PII promoter (*pAt*PII*::At*PII**_genomic_-*GFP*) and co-expressed with mCherry-tagged transit peptide of tobacco Rubisco (CD3-999 pt-rk; magenta) localizes to plastids in transiently transformed *N. benthamiana* 2 days after infiltration. (**C**) Genomic *At*PII-GFP (green) under the control of endogenous *pAt*PII (*pAt*PII*::At*PII**_genomic_-*GFP*) localizes to plastids (magenta) in stably transformed *A. thaliana*. In each row, the GFP fluorescence is shown first, the mCherry fluorescence is shown second, and the merge of both pictures is shown last. White arrows mark exemplarily *At*PII aggregates.

**Figure 2 ijms-22-12666-f002:**
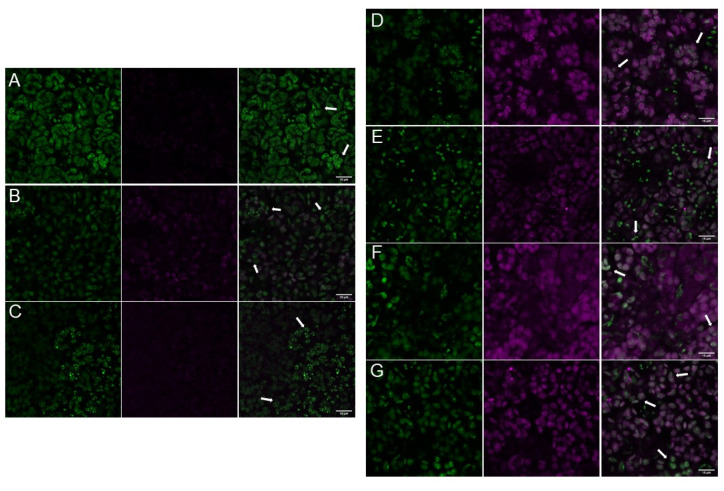
PII foci formation in *Arabidopsis thaliana* under different temperature and light regimes. Expression of genomic *At*PII-GFP under the control of the *pUBQ10* promotor (*pUBQ10::At*PII**_cDNA_-*GFP*) was analyzed in 6-day-old seedlings. Seedlings were incubated for 24 h for temperature treatment in the dark at (**A**) RT, (**B**) 8 °C, (**C**) 37 °C, and for light treatment in (**D**) blue light, (**E**) green light, (**F**) red light, and (**G**) far red light. Seedlings were fixed after incubation. In each row, the GFP fluorescence is shown first, chlorophyll autofluorescence is shown second, and the merge of both pictures is shown last. White arrows: PII foci. Scale bar: 10 µm.

**Figure 3 ijms-22-12666-f003:**
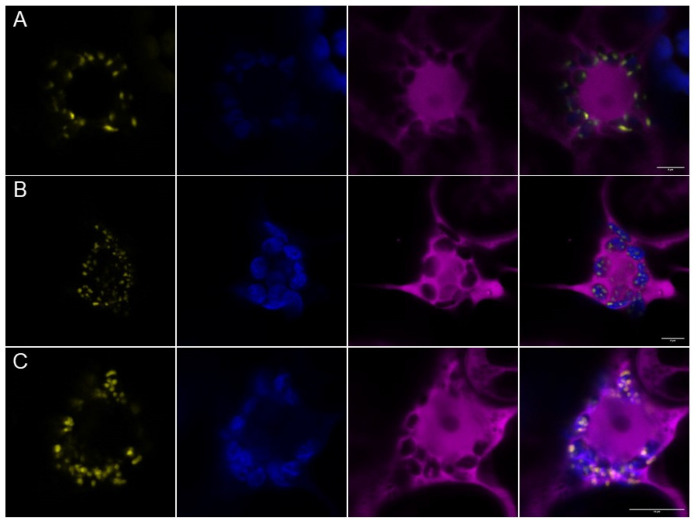
BiFC analysis of *At*PII with itself and known interactors in PII foci. *At*PII-nYFP was co-expressed with *At*PII-cYFP (**A**), *At*NAGK-cYFP (**B**), and *At*BCCP1-cYFP (**C**), respectively, under the control of *p35S* promotor using 2in1-BiFC vectors. Images were taken 3 days after transient transformation of *N. benthamiana* leaves. In each row, the YFP fluorescence (yellow) is shown first, chlorophyll autofluorescence (blue) is shown second, free RFP fluorescence as expression control (magenta) is shown third, and the merge of all pictures is shown last. Scale bar: 10 µm.

**Figure 4 ijms-22-12666-f004:**
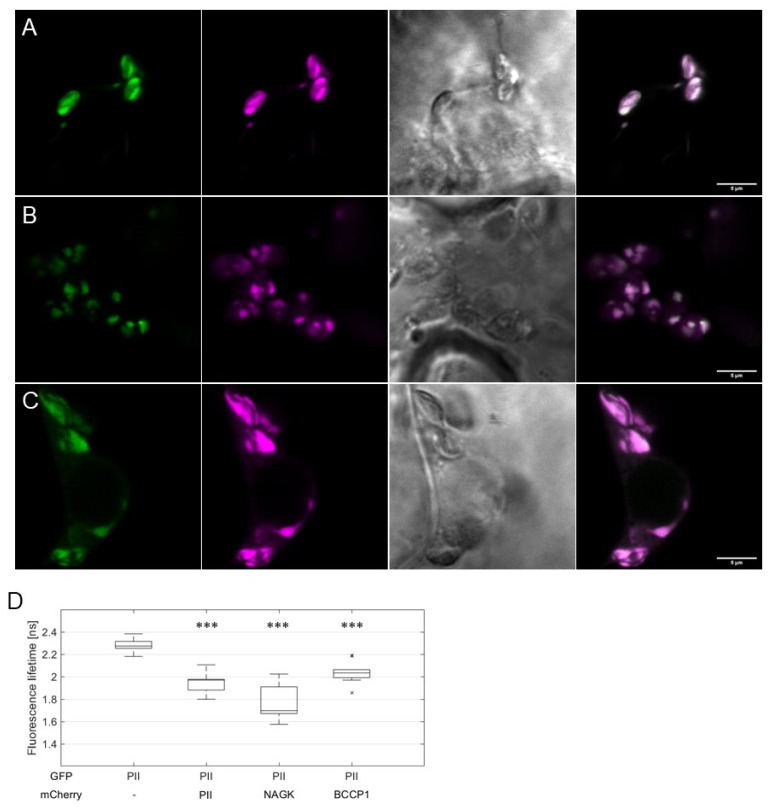
FRET/FLIM analysis of *At*PII interaction with candidate proteins in PII foci. *At*PII-GFP was co-expressed with *At*PII-mCherry (**A**), *At*NAGK-mCherry (**B**), and *At*BCCP1-mCherry (**C**), respectively, under the control of *p35S* promotor in *N. benthamiana* using 2in1 FRET vectors. Images were taken 2 days after transient transformation of *N. benthamiana*. In each row, the fluorescence of GFP is shown first, mCherry (magenta) is shown second, the brightfield image is shown third, and the merge of both fluorescence pictures is shown is shown last. Scale bar: 10 µm. (**D**) FLIM analyses of fluorescent co-localising signals in (**A**–**C**). Student’s *t*-test used for calculation of significance. Data points marked with an “x” represent statistical outliers of measurement. *** *p* < 0.001.

**Figure 5 ijms-22-12666-f005:**
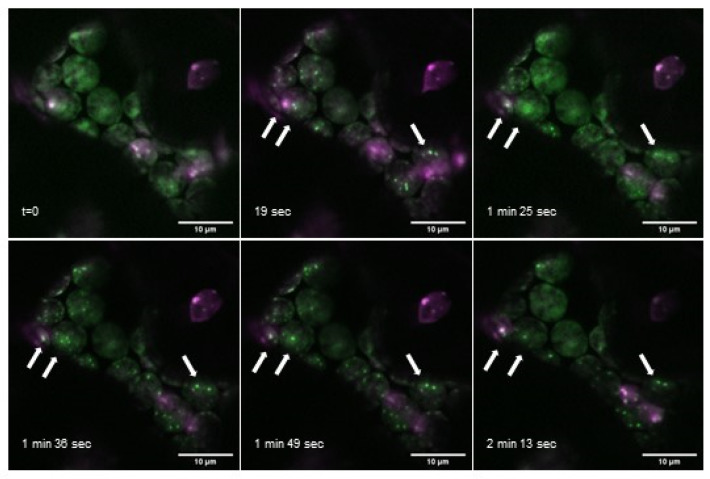
Appearance of PII foci over time. Overlay images of time series of *At*PII-GFP (green) and *At*NAGK-mCherry (magenta), both expressed under the control of *p35S,* 2 days after transient transformation of *N. benthamiana.* White arrows mark exemplarily PII foci observable over the whole time range. Scale bar: 10 µm.

**Figure 6 ijms-22-12666-f006:**
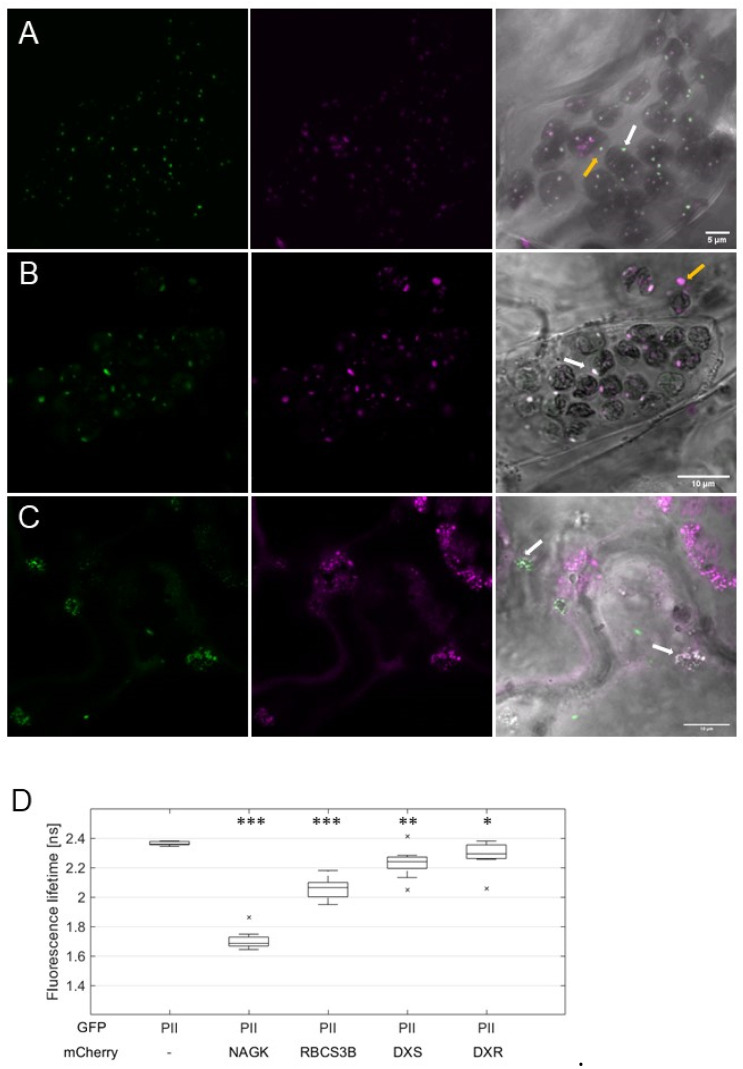
*At*PII is found in different plastidial aggregates. *At*PII-GFP was co-expressed with *At*RBCS3B-mCherry (**A**), *At*DXR-mCherry (**B**), and *At*DXS-mCherry (**C**), respectively, under the control of *p35S* in *N. benthamiana*. In each row, the GFP fluorescence is shown first, the mCherry fluorescence is shown second, and the merge of both fluorescence images with the brightfield image as background is shown last. White arrows mark exemplarily *At*PII aggregates in chloroplasts (dark and round structures in the brightfield image), orange arrows indicate extraplastidic vesicle-like structures. (**D**) FLIM analyses of fluorescent co-localizing signals in (**A**–**C**) together with *At*PII-GFP/*At*NAGK-mCherry as positive control. Student’s *t*-test used for calculation of significance. Data points marked with an “x” represent statistical outliers of measurement. * *p* < 0.05; ** *p* < 0.01; *** *p* < 0.001.

**Figure 7 ijms-22-12666-f007:**
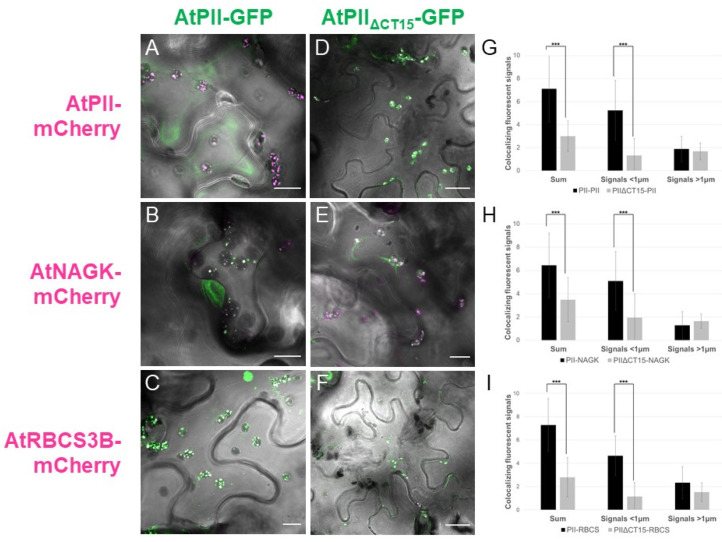
Truncation of the C-terminus of *At*PII leads to different aggregation behavior. *At*PII-GFP or *At*PIIΔCT15-GFP were co-expressed either with *At*PII-mCherry ((**A**,**D**), respectively), *At*NAGK-mCherry ((**B**,**E**)**,** respectively) or *At*RBCS3B- mCherry ((**C**,**F**), respectively), under the control of p35S in *N. benthamiana*. Each of these pictures shows the merge of GFP and mCherry fluorescence with the brightfield image as background. Size bars show 10µm except for (**D**) and (**F**) where they indicate 22 µm. In G-I, the average total number (Sum) and the number of small and large co-localizing fluorescent signals per chloroplast for the co-expressions of *At*PII-GFP/*At*PII_ΔCT15_-GFP + *At*PII-mCherry (**G**), + *At*NAGK-mCherry (**H**), and *At*RBCS3B-mCherry (**I**) are provided (*n* = 25). Student’s *t*-test used for calculation of significance. *** *p* < 0.001.

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
