# Peer review of "Arabidopsis PII Proteins Form Characteristic Foci in Chloroplasts Indicating Novel Properties in Protein Interaction and Degradation"

_ijms, 2021, doi:10.3390/ijms222312666_

Round 1

Reviewer 1 Report

It is an interesting article. I have few comments for authors mentioned below:

Abstract: add take home message at the end.

Introduction: I suggest to strengthen the hypothesis part.

Figure 7, Graphs are not clear. Provide graphs with clear and visible text

There should be a separate conclusion section

Author Response

Dear reviewer,

first of all, many thanks for your interest in our work and the valuable comments.

Point for point reply to your comments:

  1. We revised the last two sentences of the abstract in a way to provide a take home message of the study.
  2. The last paragraph of the Introduction was revised, that the hypothesis and approach of the study got (hopefully) clearer to the reader.
  3. The text in graphs of Fig. 7 are larger now.
  4.  We abstained from introducing an additional conclusion section. Therefore, the text would have been partially rewritten, with which the other reviewer seemed to be rather happy. As a compromise prospective sentences were added to the ends of some paragraphs of the Discussion to emphasize the conclusion character of this section.
  5. Additionally, extensive grammar and language editing was done.

Best regards,

Üner Kolukisaoglu

Reviewer 2 Report

In this manuscript, the authors study the Arabidopsis PII proteins form characteristic foci in chloroplasts that indicate novel protein interaction and degradation properties. In this study, to gain more insights into the function of PII proteins protein, authors investigated the interaction behavior of AtPII with candidate proteins by BiFC and FRET/FLIM in planta and with GFP/RFP traps in vitro. In the course of these studies, authors found that AtPII interacts in chloroplasts with itself and known interactors like NAGK in dot-like aggregates, which authors named as PII foci. Furthermore, in these novel protein aggregates, AtPII also interacts with yet unknown partners, which are involved in plastidic protein degradation. Further studies revealed that the C-terminal part of AtPII is crucial for the formation of PII foci. Altogether, the presented results indicate a novel mode of interaction for PII proteins with other proteins in plants, which may be a new starting point for the elucidation of physiological functions of PII proteins in plants.

The manuscript is written very well, backed by solid data. I found no major faults in this manuscript and congratulate the authors for writing an excellent manuscript. Therefore, the manuscript can be accepted in its current format.

Author Response

Dear reviewer,

we were quite happy about your interest in our work and the positive comments. As suggested extensive grammar and language editing of the text was done.

Best regrads,

Üner Kolukisaoglu